# Monkeypox and Pregnancy: Latest Updates

**DOI:** 10.3390/v14112520

**Published:** 2022-11-14

**Authors:** Alexandre Cuérel, Guillaume Favre, Manon Vouga, Léo Pomar

**Affiliations:** 1Department Woman-Mother-Child, Lausanne University Hospital and University of Lausanne, 1011 Lausanne, Switzerland; 2School of Health Sciences (HESAV), HES-SO University of Applied Sciences and Arts Western Switzerland, 1011 Lausanne, Switzerland

**Keywords:** monkeypox, pregnancy, vertical transmission, vaccine, miscarriage, orthopox, smallpox, delivery

## Abstract

Monkeypox virus (MPXV) has emerged as a threatening zoonosis. Its spread around the world has been growing fast over the last 2 years, particularly in 2022. The reasons for this sudden spread are probably multifactorial. The R0 values of the two MPXV clades are rather low, and a massive pandemic is considered unlikely, although the increase in the number of single-nucleotide polymorphisms found in the 2022 MPXV strain could indicate an accelerated human adaptation. Very little is known about the risks of an infection during pregnancy for both the mother and the fetus. Further observations must be made to create clear, adapted, evidence-based guidelines. This article summarizes the current knowledge about MPXV infections and similar pregnancy virus infections.

## 1. Epidemiology

Monkeypox virus (MPXV) was isolated for the first time from a monkey at the end of 1950s in Denmark [1], and the first human infection was reported in 1970. The virus pathogen was identified in a nine-month-old child in the Democratic Republic of Congo, who presented with a fever, followed by a centrifugal rash and enlarged cervical lymph nodes [2,3]. Since then, other cases caused by MPXV have been reported in West Africa, and the virus is currently considered as an endemic virus in the region [4], with several outbreaks described over the last 50 years [5], which were observed in Cameroun [4], Gabon [6], the Democratic Republic of Congo [7], the Republic of the Congo [8], Sierra Leone [9], Nigeria [10], Sudan [11] and Liberia. The first description of MPXV outside Africa was reported in 2003 in the USA [12]. After that, cases were reported in 2018 in Israel [13,14], in 2019 in Singapore [15] and in 2021 in the UK [16], where the virus was imported from Ghana and Nigeria. In 2022, we have been faced with a massive spread of the disease. Over 40 countries have identified the virus, and the cases have grown rapidly in the months leading up to August 2022. On 30 October 2022, the World Health Organization estimated a total 76,871 MPXV cases and 36 fatalities around the globe [17].

The last epidemiologic updates show that the peak level of infections occurred at the beginning of August 2022, and the numbers are now decreasing rapidly [18]. The vast majority of the infections are now identified on the American continent, whereas Europe was predominant at the beginning of the 2022 outbreak [17].

Two main clades have been identified, including the former Congo Basin (CB) clade, now called clade I, and the former West Africa (WA) clade, now called clade II [19]. The first one appears to be more virulent than the second one, with an R0 of 0.6 to 1.0 for the CB clade [20]. The R0 of the WA clade still has yet to be determined but is probably lower. The high virulence of the CB clade indicates that its inter-human transmission is not negligible [21]. The CB clade is also more lethal, with a case fatality rate of 11% versus < 1% for the WA clade [22]. More recently, a third clade has been described. It was observed for the first time during the 2017–2019 Nigeria, Singapore, UK and USA outbreaks and has caused the rapid worldwide spread observed in 2022 [23]. Luna and et al. subdivided clade 3 into different lineages, analyzing 337 monkeypox genomes. They identified one lineage in particular, called B1, which appeared in Europe between the end of 2021 and beginning of 2022 and has caused most of the infections worldwide in the 2022 outbreak [24]. Happi and et al. proposed the names of clade IIb instead of clade III and clade IIa instead of clade II, because both come from the same former West African clade [25].

The exact reservoir of MPXY has not been identified yet. Many mammalians are suspected to be natural hosts, and MPXV has been identified in monkeys, mice, rats, porcupines, woodchucks, squirrels, etc. [26,27,28,29]. Despite its name, the evidence suggest that monkeys are not the main hosts. The transmission between animals and humans remains unclear, but proximity to an infected animal (through direct contact or fluid contact) seems to be the main means of transmission [30,31]. Inter-human transmission through respiratory droplets has also been observed in the case of oropharyngeal lesions, infected fluids and direct contact [32]. The recent spread has mostly been associated with sexual transmission, affecting men having sex with men and those who have multiple partners in particular through direct contact with the lesions and fluids, even if they do not represent the only population at risk [33]. Young people who have not been smallpox-vaccinated are more at risk of infection, and studies reveal that men are more commonly contaminated, with a female–male ratio of 1–2.5 [34]. New studies show that the proportion of affected males may be even higher.

The 2022 outbreak of MPXV appears to be the largest observed thus far. The number of countries affected and the worldwide geographic distribution are unprecedented [35]. The phenomenon might be explained by several characteristics, including: (1) increasing direct contact between the West Africa population and the wildlife reservoirs, caused by deforestation [36]; (2) the possible increased inter-human transmission of the virus, as although the WA clade is responsible for most of the infections, the increase in the R0 of the CB MPXV clade is concerning and suggests a potential ability to mutate and adapt to human hosts [37,38]; (3) the lower proportion of people immunized against smallpox, with the vaccination program stopped in 1980 [37]; and (4) the technical progress and the growing interest in the disease, enabling a better diagnosis [39].

## 2. Virus Characteristics

The MPXV (Figure 1) belongs to the Orthopoxvirus genus, which belongs to the family of Poxviridae [40]. It is a linear DNA virus of approximately 197 kB. Along with the Variola virus, cowpox virus and Vaccinia virus are the only members of the Orthopoxvirus genus that are able to infect humans [41]. The evolution of Orthopoxviruses depends on the recombination of (that takes place during host infection) [42] and variation in the genome (loosing and gaining DNA) [43]. MPXV’s long DNA provides it with a good stability and, therefore, a lower ability to mutate [44]. Nevertheless, the 2022 MPXV B.1 clade has been sequenced and has revealed a high number of single-nucleotide polymorphisms, with possible increased capacities for inter-human transmission, mutation, and adaption [37,45]. It is noteworthy that the WA clade has a longer genome than the CB clade, which might contribute to its lower pathogenicity [31].

## 3. Pathophysiology

Like all viruses, MPXVs are intracellular pathogens, meaning that their survival depends on their host [46]. This type of virus has the ability to encode proteins that interfere with the immune response of the cells and targets different cell pathways, such as apoptosis [47,48].

The MPXV has been identified in several mammals. Studies investigating the MPXV tissue tropism conducted on immunodeficient mice revealed a particular tropism for ovarian tissues, which may have impacts in the field of gynecology and obstetrics. However, a similar tropism for human ovary tissues has not been proved.

As demonstrated, the virus can be transmitted through different means, including direct contact with skin lesions, respiratory droplets from saliva and, possibly, sexual intercourse, as MPXV has also been identified in sperm [49,50]. Initially, the virus multiplies at the inoculation site (oropharyngeal, skin and anogenital areas, depending on the transmission mode) before draining into the closest lymph nodes. At this point, a viremia can be detected (called primary viremia, which lasts until the end of the incubation period and is not associated with infectivity). This is then followed by the secondary viremia, which is characterized by the important replication of the virus responsible of the main symptoms of the prodromal phase (lymphadenopathy, fever, myalgia). At this point, the virus increasingly disseminates throughout the body and reaches the skin and mucosa, where lesions emerge, typically 1 to 3 days after the fever appears [51].

## 4. Clinical Manifestations

The period of incubation ranges from 7 to 14 days. At this point, the subject remains asymptomatic. It is followed by the prodromal phase, where the infected subject starts to become contagious, and lasts for 1–4 days. The classical symptoms at this point are fever, fatigue, myalgia, headaches and lymphadenopathy (1 to 4 cm maxillary, tender cervical and inguinal lymph nodes) [52,53]. Lymphadenopathy is the main symptom distinguishing chickenpox from monkeypox infection, as they are not found in chickenpox-infected people. The prodromal phase is followed by the rash phase (Figure 2), with a decline in the fever and the onset of cutaneous macular lesions. These evolve to popular, vesicular and, finally, pustular lesions. Classically, the rash starts on the face and spreads centrifugally [54]. New studies show a specific tropism for the genital area [55]. Potential life-threatening complications can occur, such as encephalitis and diarrhea causing dehydration, as well as secondary bacterial infections, such as bronchopneumonia or pharyngeal abscesses. Ocular involvement causing corneal damage and definitive vision loss [22] have been observed as well. Finally, like smallpox, permanent skin lesions are frequent.

## 5. Pregnancy Involvement

Very few studies have reported MPXV infection among pregnant women, and few data have been collected to date. A possible reason for this might be the presence of the virus in deep rural regions of Africa with a limited technical capacity [36], and another reason might be that it affects mostly young men [33], and infection among pregnant women is rare.

Mbala and et al. reported the outcomes of four pregnant women in a study conducted in the Democratic Republic of Congo based on 222 symptomatic MPXV-positive patients from 2007 to 2011. Two of them contracted MPXV infection during the first trimester and presented with a spontaneous abortion 14 to 24 days after the onset of fever. These two patients developed a moderate to severe disease. The embryos were not analyzed, and definite conclusions are difficult to draw. The third patient became symptomatic at 14 weeks of gestation and gave birth to a healthy newborn at term. The fourth patient contracted MPXV at 18 WG, and the pregnancy resulted in a late spontaneous abortion 21 days after the beginning of the fever. The examination of the fetus resulted in the identification of maculopapillary lesions on the extensive parts of the body, associated with hydrops fetalis and hepatomegaly. No malformations were described. The analysis of the placenta revealed diffuse images of hemorrhages on the maternal side. MPXV DNA was detected by the PCR technique both in the placenta (2.4 × 10^7^ copies) and in the fetus (1.6 × 10^3^ copies), suggesting materno-fetal transmission. Interestingly, a rapid rise in viremia was observed from day 21 to day 23 after the onset of fever. At this point, the mother felt fewer movements, and the death of the fetus was revealed [56,57].

Lately, the identification of perinatal MPXV infection was reported among a familial cluster. A healthy newborn developed a vesico-pustular rash after 9 days of life, which was rapidly complicated by respiratory failure requiring invasive ventilation and a 4-week stay in the neonatal intensive care unit. Retrospective anamnesis revealed that both the father and the mother presented with a rash, respectively, at 9 days before the birth and 4 days after the birth. The mother and the newborn were positive for MPXV, and a co-infection with an adenovirus was also diagnosed in the newborn. Unfortunately, the timing of the newborn’s infection is difficult to establish. The infection could have been contracted in utero or at the time of delivery, but postnatal acquisition is possible [58]. Similarly, the presence of a co-infection makes it difficult to assess the severity of the MPXV infection [59].

Although the data are limited, these cases suggest the potential for vertical transmission of MPXV, either transplacental or at the time of delivery, similar to other well-known diseases, such as the Varicella zoster virus or Herpes Simplex viruses [60,61]. However, more studies must be performed to reveal the risk factors and impacts on the fetus. Assuming that the vertical transmission is comparable to that of most vertical transmissible viruses, if the mother is infected, the risk of transmission might depend on the number of weeks of gestation. As with other viruses, the likelihood of transmission may increase as the pregnancy advances, with a higher risk of transmission during the third trimester than the first trimester and a maximized risk at the time of delivery, particularly in the presence of active maternal lesions [62]. To date, the proportion of vertical transmissions during pregnancy remains unknown. To establish the risk of transmission, prospective cohort studies should be conducted, as in the case of the Zika infections in 2017 [63]. A new European prospective cohort study has just started and intends to gather information about the vertical transmission of MPXV [64]. The effect on the placenta has not been described, and further anatomopathological studies must be conducted. The mechanism of MPXV vertical transmission remains unknown.

Although the few cases reports that exist are reassuring, caution is also warranted with respect to maternal complications. Indeed, before vaccination, smallpox infection among pregnant women is associated with a high rate of fatality, particularly during the third trimester. In a retrospective historical meta-analysis, the overall case fatality was estimated to range from 31.4 to 37.1%, and the proportion of miscarriages or premature births among infected women was determined to range from 36.5–43.2% [65,66].

## 6. Diagnosis

Anamnesis is central to the identification of potential exposed patients. First of all, the patients at risk have to be identified (travelers coming from an MPXV-endemic or highly affected region, who have been in contact with an infected person or exposed to a wild mammal [67]). Then, the physician must be trained to recognize the different symptoms of the disease. If mucosa or skin lesions are visible, a PCR swab test can be performed on the lesions. Otherwise, an oropharyngeal swab can be conducted, along with a PCR test of the body fluids (blood, vaginal discharge, urine) [68]. To rule out the different infections having similar clinical aspects, chickenpox, syphilis, herpes virus and haemophilus dureii should be tested for at the same time.

MPXV can be found in the body fluids (saliva, urine, sperm) [68] and probably in vaginal swabs, but this still has yet to be proved. It can be identified through active skin wounds too. Therefore, close contact is the main mode of transmission, as well as sexual contact. Its transmission through saliva droplets still has yet to be proved.

As shown by H. Adler and et al., the largest number of viral copies can be found in the ulcerated lesions of the skin and, therefore, the best way to detect the infection early is a skin swab, followed by a swab of the upper respiratory tract [68]. The WHO also recommends diagnosis by PCR analysis of the skin lesions [5].

## 7. Treatment

An antiviral treatment should be considered for severely ill patients, such as Tecovirimat. It is FDA-approved for the treatment of smallpox, and its use for other Orthopoxviruses (including MPXV) under the Investigational New Drug protocol is accepted in the USA. Its use for the treatment of MPXV is also validated by the European Medicine Agency. Still, its use has not been authorized for pregnant patients as of yet [60]. However, animal research has observed no embryo toxic/teratogenic effect of Tecovirimat [69]. The risks and benefits of its use should be exposed, and treatment might be considered for severely ill pregnant women. In opposition, Cidofovir and Brincidofovir are contraindicated, as animal studies have demonstrated evidence of teratogenicity [70].

Immunoglobulins, such as vaccinia immunoglobulin, can be used as preventive treatments for immunocompromised patients, but they have not yet been validated for pregnant women [71].

Exposed subjects, both symptomatic and asymptomatic, should be PCR-tested. In the case of a positive test, hospitalization should be recommended. Vaccination should be advised for asymptomatic patients if exposure occurred less than 14 days ago [72].

## 8. Prevention

Studies show that most transmissions occur through direct contact or contact with body liquids. Hygiene measures must be the primary prevention strategy.

All smallpox vaccines offer good cross-protection against MPXV infection. Smallpox vaccination drastically increased the survival of smallpox-infected people; the fatality rate fell by 1–2% for men and non-pregnant women and 26% for pregnant women. These numbers indicate the important protection provided by vaccination against this class of viruses. Concerns exist regarding the use of second-generation vaccines based on the administration of the live vaccinia virus (e.g., ACAM2000) and their potential secondary effects. These are contraindicated in the case of pregnant women.

The MVA-BN vaccine was tested on 300 pregnant women, with no secondary effect reported on the pregnancies [73].

Third-generation smallpox vaccine should thus be recommended to all women who are planning a pregnancy, and if vaccination is authorized for a pregnant woman, it should be recommended in the four days following her exposure to MPXV. This recommendation is based on the observation made by Fenner (1988) on smallpox, showing that early vaccination after exposure increases the survival of males and non-pregnant females [66,74,75].

The French health society HAS recommends third-generation vaccination for women at risk, such as women with multiple partners or those sharing a living space with people at risk (prostitutes, MSM with multiple partners) [76].

Considering the mode of transmission, facemask protection, gloves, glasses and protective suits should be worn during a medical interaction with an infection-suspected patient. The patient should be isolated at home or at the hospital, depending on the severity of the symptoms. A negative pressure room should be considered, as long as the possibility of aerosol transmission has been excluded. A negative oropharyngeal PCR swab or skin PCR swab should rule out the active infection and enable the release from confinement.

## 9. Fetal Monitoring

In cases of suspected or confirmed infection during pregnancy, the patient should be referred to a tertiary center for referral imaging [45]. At present, specific recommendations for the antenatal imaging of monkeypox-exposed fetuses are lacking, and the following recommendations are based on the ISUOG guidelines for monitoring infections during pregnancy [77].

Depending on the trimester, different examinations should be performed. For a first-trimester pregnancy, an echography should confirm the viability. Then, the first-trimester echography followed by an early morphologic echography (enabling the observation of early signs of infection) and then monthly referral ultrasound should be proposed to follow the growth of the fetus and to diagnose possible late-onset manifestations during the fetal development. These echography exams should include the amniotic volume, the estimation of the fetal growth, MCA and umbilical cord Doppler measure, an evaluation of the placenta, and the search for general and specific fetal infectious signs, including hydrops or effusions, hyperechogenic bowels, hepato-splenomegaly and brain and eye structural anomalies. Furthermore, one cannot exclude the possibility that congenital MPXV acquired early in pregnancy could lead to embryopathies, including further birth defects. This echography should be repeated once per month for the remaining time of the pregnancy. An additional MRI of the fetal brain could be discussed between 30 and 34 weeks of gestation, depending on the conditions and results of the ultrasound examinations and the local possibilities [78] (Figure 3).

For second- and third-trimester infections, the follow-up should be the same as that of the first-trimester infection, with a referral ultrasound performed monthly. According to the study by Mbala et al., a close follow-up (on a weekly basis) could be added after the acute phase of the maternal infection, with vitality, Doppler and amniotic fluid assessments, due to the risk of intra-uterine fetal demise [57].

If an echography arouses suspicion of fetal infection over 18–21 weeks of gestation, an amniocentesis could be proposed to confirm the presence of the virus in the amniotic fluid [56,79]. Nevertheless, the sensitivity and specificity of MPXV PCR using amniotic fluid samples are not known. The benefit of such invasive testing should be balanced with the risk of complications, particularly the risk of miscarriage (0.2–1%).

## 10. Timing and Mode of Delivery

Similar to other infections, it is very likely that most MPXV infections will not affect the fetus. Therefore, delivery should only be recommended to severely ill pregnant women [45]. A multidisciplinary team should discuss the different possibilities.

Cesarean should not be proposed to all infected pregnant women. The current data is insufficient to show the benefit of C-section over vaginal delivery. In the presence of genital skin lesions, a discussion should be conducted with the patient to inform her about the risks, and the decision should be made on a case-by-case basis [80].

## 11. Neonatal Care

First of all, the newborn should be tested (oropharyngeal PCR swab, skin lesion swabs and placenta swab), combined with an umbilical cord PCR blood test. If the newborn is negative, in order to prevent a neonatal infection, contact should be avoided if the mother presents with an active infection [70,80]. As Adler et al. demonstrated, the MPXV can be found weeks after the onset of fever in the oropharyngeal swabs and skin lesion swabs [68]. Two negative PCR tests (skin and oropharyngeal swabs) should be obtained before allowing the mother and the baby to be reunited. If the baby is infected, there is no need for the separation of the mother and the child.

The presence of the MPXV in breast milk has not yet been demonstrated. In the case of both the baby and mother being infected, breastfeeding should be allowed. In the case of an MPXV-positive mother and -negative baby, breastfeeding should be avoided until the mother has PCR-tested twice as negative. The milk (obtained with a breast pump) should only be given to the newborn if it has PCR-tested as negative.

In the case of an MPXV-positive mother willing to breastfeed her MPXV-negative newborn, healthcare professionals should first exclude the presence of skin lesions around the nipples and breasts and then advise the mother to wear a facemask during breastfeeding [45].

Finally, third-generation smallpox vaccination should be discussed for the newborn if the mother is positive [45].

## 12. Conclusions

MPXV became a medical topic of interest suddenly, mainly because of its rapid worldwide spread during recent years and months. Recent updates suggest that the end of the 2022 outbreak is near. The number of reported infected pregnant women remains very low, and the available data are almost inexistent. According to the recent discoveries, this virus could represent a potential threat for pregnant women and fetuses, even if the amount of information is too limited to draw conclusions at present. In the case of a future re-emergence of the MPXV outbreak, the guidelines will remain useful and applicable. The specific risks and numbers during pregnancy still have yet to be described. In the meantime, we recommend a close follow-up, allowing for the observation of early virus-related complications and treatment responses.

## Figures and Tables

**Figure 1 viruses-14-02520-f001:**
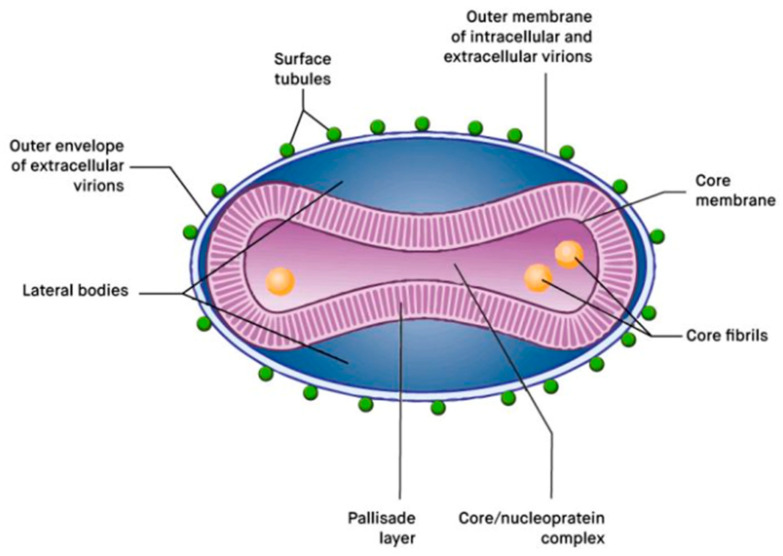
Monkeypox virus. Schematic representation of a poxvirus particle. Adapted from *Principles of Molecular Virology*, 6th Edition (p. 46), by Alan J. Cann, 2016, UK: Elsevier. Copyright 2016 by Elsevier. License number for the reuse of this figure: 5415960043412.

**Figure 2 viruses-14-02520-f002:**
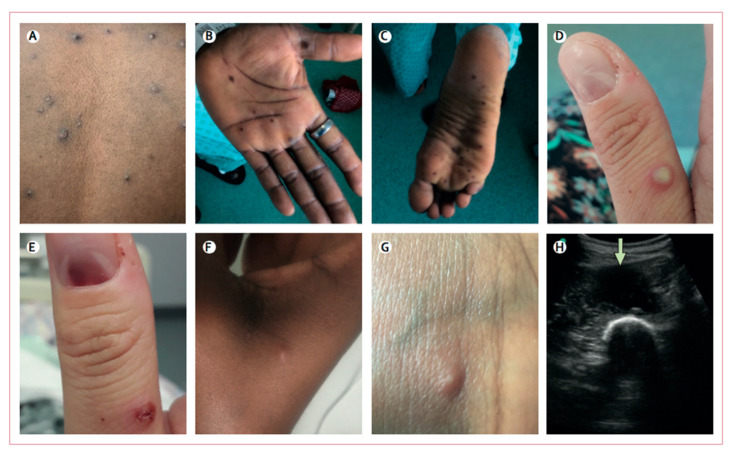
Skin and soft tissue manifestations of monkeypox. Skin and soft tissue features included: (**A**,**D**) vesicular or pustular lesions; (**B**,**C**) macular lesions involving the palms and soles; (**D**,**E**) a sub-ungual lesion; (**F**,**G**) more subtle papules and smaller vesicles; (**H**) and a deep abscess (arrow, image obtained during ultrasound-guided drainage). From: Adler H, Wingfield T, Price NM: Clinical Features and Management of Human Monkeypox: A Retrospective Observational Study, in the UK *Lancet Infect Dis* 2022; 22: 1153-62. Copyright 2022 by Elsevier. License number for the reuse of this figure: 5414980739216.

**Figure 3 viruses-14-02520-f003:**
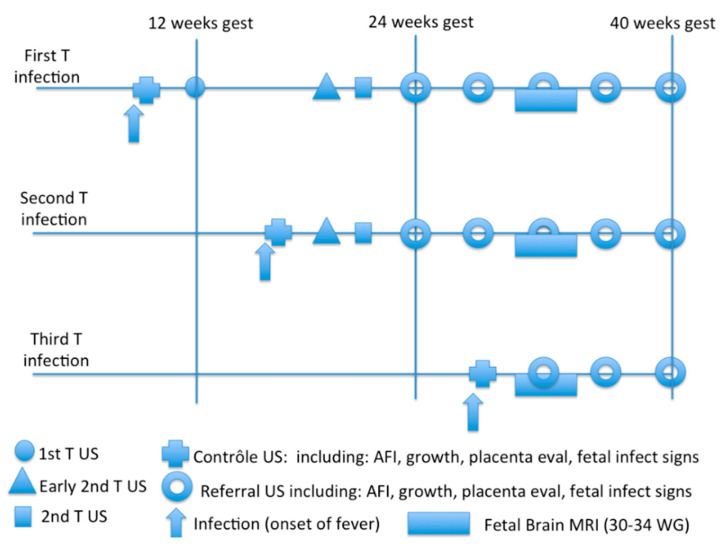
Proposed follow-up in the case of MXPV infection during pregnancy.

## Data Availability

Not applicable.

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
