# Peer review of "Monkeypox and Pregnancy: Latest Updates"

_viruses, 2022, doi:10.3390/v14112520_

Round 1

Reviewer 1 Report

Monkeypox and pregnancy: latest updates.

This is a very complete and helpful review article for obstetrician-gynecologists, however, its contribution is minimal compared to other recently published reviews and several mentioned as references in this paper.

There are several errors in the indications of the references, in the way of indicating them in the text and in the non-coincidence between the one indicated in the text and the true one, let's see:

- In line 26 it says 34, it should be 3.4.

- In line 43, 26 appears, after 22 and before 23.

- In line 57 they are misplaced

- Line 60 is missing a comma between 34 and 35.

- Line 75 is missing a comma between 43 and 44.

- In line 120 it should be 62-64.

- Between lines 144 and 159 refer to the article by Mbala et al, but reference 70 is another.

- Line 191 is missing a comma between 77 and 78.

- In line 206 (Adler and al) a reference is missing, same in line 208 (WHO).

- In line 242 the reference must be 87-89.

- Between lines 276 and 278 it refers to Mbala at al and now it is with reference 94, this coincides and not so in the previous mention with reference 70.

- Line 300 is missing a comma between 98 and 99.

Figure 1 and Figure 2 are not marked in the text.

The content of lines 289 and 290 on lung maturation and neuroprotection should be deleted, as this has not been investigated with this disease.

Author Response

We thank the reviewer for this positive comment. We recognise that several landmark papers on the subject have come out in recent months. These papers are mentioned in our review, which will find its place in this special issue on emerging viruses and adverse pregnancy outcomes.

We thank the reviewer for their careful reading and have made the corrections required. The entire bibliography has been revised to address these remarks.

Figures 1 and 2 are now mentioned in the text (lines 80 and 123).

According to this advice, these recommendations have been deleted.

Reviewer 2 Report

This article is written well and is good for publication in the journal. However some points to be addressed before its publication, which I have listed below:

1. I have suggested some changes in the file, so update the file accordingly.

2. I suggest reading and incorporating some infromation from the article-Possibility of vertical transmission of the human monkeypox virus.Int J Surg. 10.1016/j.ijsu.2022.106832

3. References are not as per the standard format of the journal. Update accordingly.

Rest is ok.

Author Response

We thank the reviewer for this positive comment and for their help to improve our manuscript. However, we have repeatedly asked editorial support for access to the file mentioned, but it seems that they have not received your file with your change requests. 

This reference has been added in the manuscript. 

The references have been updated according to the journal format.

Round 2

Reviewer 1 Report

Corrections were made, thank.